# Cervicovaginal Human Papillomavirus Genomes, Microbiota Composition and Cytokine Concentrations in South African Adolescents

**DOI:** 10.3390/v15030758

**Published:** 2023-03-15

**Authors:** Anna-Ursula Happel, Christina Balle, Enock Havyarimana, Bryan Brown, Brandon S. Maust, Colin Feng, Byung H. Yi, Katherine Gill, Linda-Gail Bekker, Jo-Ann S. Passmore, Heather B. Jaspan, Arvind Varsani

**Affiliations:** 1Department of Pathology, University of Cape Town, Anzio Road, Observatory, Cape Town 7925, South Africa; 2Institute of Infectious Disease and Molecular Medicine, University of Cape Town, Anzio Road, Observatory, Cape Town 7925, South Africa; 3Seattle Children’s Research Institute, 307 Westlake Ave. N, Seattle, WA 98109, USA; 4Department of Pediatrics, University of Washington, 1510 San Juan Road NE, Seattle, WA 98195, USA; 5Desmond Tutu Health Foundation, 3 Woodlands Rd, Woodstock, Cape Town 7915, South Africa; 6National Health Laboratory Service, Observatory, Cape Town 7925, South Africa; 7NRF-DST Center of Excellence in HIV Prevention, Centre for the AIDS Programme of Research in South Africa, 719 Umbilo Road, Congella, Durban 4013, South Africa; 8Department of Global Health, University of Washington, 1959 NE Pacific St., Seattle, WA 98195, USA; 9The Biodesign Center for Fundamental and Applied Microbiomics, Center for Evolution and Medicine and School of Life Sciences, Arizona State University, Tempe, AZ 85287, USA; 10Structural Biology Research Unit, Department of Integrative Biomedical Sciences, University of Cape Town, Observatory, Cape Town 7925, South Africa

**Keywords:** HPV, *Alphapapillomavirus*, *Gammapapillomavirus*, whole genome, DNA viruses, vaginal virome, genital inflammation, microbiome, Sub-Saharan Africa, women’s health

## Abstract

The interaction between cervicovaginal virome, bacteriome and genital inflammation has not been extensively investigated. We assessed the vaginal DNA virome from 33 South African adolescents (15–19 years old) using shotgun DNA sequencing of purified virions. We present analyses of eukaryote-infecting DNA viruses, with a focus on human papillomavirus (HPV) genomes and relate these to the vaginal bacterial microbiota (assessed by 16S rRNA gene sequencing) and cytokines (assessed by Luminex). The DNA virome included single-stranded (*Anelloviridae*, *Genomoviridae*) and double-stranded DNA viruses (*Adenoviridae*, *Alloherpesviridae*, *Herpesviridae*, *Marseilleviridae*, *Mimiviridae*, *Polyomaviridae*, *Poxviridae*). We identified 110 unique, complete HPV genomes within two genera (*Alphapapillomavirus* and *Gammapapillomavirus*) representing 40 HPV types and 12 species. Of the 40 HPV types identified, 35 showed positive co-infection patterns with at least one other type, mainly HPV-16. HPV-35, a high-risk genotype currently not targeted by available vaccines, was the most prevalent HPV type identified in this cohort. Bacterial taxa commonly associated with bacterial vaginosis also correlated with the presence of HPV. Bacterial vaginosis, rather than HPV, was associated with increased genital inflammation. This study lays the foundation for future work characterizing the vaginal virome and its role in women’s health.

## 1. Introduction

Various eukaryote-infecting DNA viruses have been identified in the genital tracts of generally healthy, asymptomatic women of reproductive age by shotgun metagenomic sequencing, including double-stranded (ds) DNA viruses (families *Adenoviridae*, *Herpesviridae*, *Papillomaviridae* and *Polyomaviridae*) and single-stranded (ss) DNA viruses (families *Anelloviridae*) [1,2]. Additionally, novel dsDNA viruses that share sequence similarity to those in the *Alloherpesviridae, Iridoviridae, Marseilleviridae, Mimiviridae, Phycodnaviridae* and *Poxviridae* families have been identified in the vaginal samples of pregnant people, and those with reproductive disorders [2,3,4,5]*.*

To date, more than 220 human papillomavirus (HPV) types have been identified, including at least 50 types that preferentially infect the genital mucosa [6]. In the Human Microbiome Project (HMP) that included healthy, asymptomatic women from North America, viruses of the genus *Alphapapillomavirus* were the most common DNA viruses detected in the lower reproductive tract, with 38% of the participants being infected with at least one *Alphapapillomavirus*. Longitudinal sampling showed the presence of many HPVs over multiple time points, suggesting that up to 50% of these papillomaviruses established productive infections [1]. Similarly, papillomaviruses were the most frequently detected viruses in women undergoing in vitro fertilization [5] and in women with human immunodeficiency virus (HIV) [7], confirming that papillomaviruses are highly prevalent in the human female genital tract (FGT), irrespective of the population studied.

High-risk HPV types have been well described to cause high-grade cervical intraepithelial neoplasia (CIN) and cancer [8], and more recently, HPV infections have been linked to increased HIV acquisition risk in sub-Saharan African women [9]. Given that young women in Sub-Saharan Africa are at the highest risk of HIV acquisition [10], and vaginal microbiota composition and genital tract inflammation have been linked to HIV risk [11,12], it is important to evaluate the interactions between HPV, cervicovaginal microbiota and inflammation in this age group. Further, although infections with low-risk HPV types do not necessarily result in clinically adverse health outcomes [8], viruses that do not cause obvious disease yet establish chronic infections have been shown to influence immunity in the gut [13]. Conversely, gut microbiota has been suggested to regulate viral infections [14]. Similar processes are likely to occur in the mucosal environment of the FGT, yet the interactions between vaginal virome, bacteriome and immunity have not been well described yet.

Bacterial vaginosis (BV) is a vaginal condition in women of reproductive age where the optimal microbiota dominated by *Lactobacillus* spp. is replaced by a range of diverse anaerobes [15]. Meta-analyses have shown an association between BV and incident HPV infection and HPV persistence [16]. Similarly, these meta-analyses showed that women with vaginal microbiota dominated by non-*Lactobacillus* spp. or *Lactobacillus iners* had up to five times higher odds of being infected with any HPV compared with women with vaginal microbiomes dominated by *L. crispatus* [17]. In these studies, HPV types were detected using commercially available HPV typing arrays that tested for the presence of several pre-specified types. These HPV typing approaches are limited by the restricted number of HPV genotypes included in commercial arrays.

Here, we used an untargeted approach to characterize eukaryotic DNA viruses in the FGT of South African adolescents to capture the FGT virus landscape in this population. Our emphasis lies on HPV genotypes, including their co-infection patterns and association with cervicovaginal bacteriomes and cytokines.

## 2. Materials and Methods

### 2.1. Study Cohort

The participants of this sub-study were recruited through a parent study, UChoose (clinicaltrials.gov/NCT02404038), which was conducted between 2015–2017 and enrolled cis-gender females aged 15–19 years. None of the participants had received an HPV vaccine, as the roll out of HPV vaccinations in the public sector in South Africa only commenced in 2014 and targeted younger girls [18]. The study was conducted in full compliance with South African Good Clinical Practice (SA-GCP), ICH76 GCP and ICMJE guidelines and approval for the study was obtained from the Human Research Ethics Committee at the University of Cape Town (HREC 801/2014). Eligibility criteria and study design have previously been described in detail [19,20,21]. Briefly, 130 non-pregnant HIV-seronegative adolescent girls were enrolled at the Desmond Tutu Health Foundation (DTHF) Youth Centre in Masiphumelele, Cape Town, South Africa, into a randomized study of injectable hormonal contraception (norethisterone enanthate, NET-EN), combined contraceptive intravaginal ring (CCVR, NuvaRing^®^; MSD Pty Ltd., Johannesburg, South Africa), and combined oral contraceptive pills (COC, Triphasil^®^ or Nordette^®^). The participants were seen at enrollment, 4 months, and 8 months. For this exploratory viral metagenome analysis, we included single time point samples from 33 participants, with samples from two longitudinal time points included for participants who experienced a change in BV status throughout the study.

### 2.2. Sample Collection

At all study visits, rapid HIV and pregnancy testing were performed, and if positive, the participant was counselled and referred for management, and no further mucosal samples were collected. Detailed interviewer-assisted questionnaires assessing medical/health history, sexual behavior, menstrual cycle, contraceptive use, intravaginal practices, and antibiotic use were completed at all visits. Venous blood was obtained for herpes simplex virus 2 (HSV-2) serological testing. Cervicovaginal secretions were collected using a disposable menstrual cup (Softcup^®^) placed over the cervix for half an hour to measure genital cytokines by Luminex, and a clinician collected a vulvovaginal swab for sexually transmitted infection (STI) testing (*Chlamydia trachomatis, Neisseria gonorrhoeae, Trichomonas vaginalis, Mycoplasma genitalium*) by multiplex PCR [22], BV testing (Gram staining and Nugent scoring; BV negative (Nugent 0–3), intermediate (Nugent 4–6) or positive (Nugent 7–10)), and pH measurement using color-fixed indicator strips (Macherey-Nagel, Düren, Germany). A clinician-collected vaginal lateral wall swab was collected for bacterial 16S rRNA gene sequencing and bacterial and viral metagenomic analysis. No samples were collected during menstruation. Upon arrival at the laboratory, the Softcup^®^ fluid and vaginal swabs were stored at −80 °C until further processing.

### 2.3. Cytokine Measurement

The Th17 cytokine panel (Bio-Rad Laboratories, Hercules, CA, USA), included 13 cytokines: interleukin (IL)-1β, IL-6, tumor necrosis factor (TNF)-α, IL-23, IL-33, IL-17A, IL-17F, IL-21, IL-22, IL-25, IL-31, interferon (IFN)-γ, and soluble CD40 ligand. Specimens from six participants were run across all plates (inter-plate controls), and samples from six participants were duplicated on each set of plates (intraplate controls) for quality control measures. Spearman’s rank test was used to measure intra-assay and inter-assay correlation coefficients to determine assay reliability and reproducibility. Data were collected using a Bio-plex Suspension Array reader, as described before [20,21].

### 2.4. Extraction of Bacterial Nucleic Acids and 16S rRNA Gene Sequencing

DNA extraction followed by amplification, sequencing and bioinformatics analysis of the V4 region of the 16S rRNA gene using 515F and 806R primers from vaginal lateral wall swabs were described previously in detail [20]. Amplicons from 96 samples were pooled in equimolar amounts, and the libraries were sequenced on the Illumina MiSeq platform (300-bp paired-end) with v3 chemistry. The raw 16S rRNA gene amplicon sequences from this study are available at https://www.ebi.ac.uk/ (accessed on 15 December 2022) under project number PRJEB30774.

### 2.5. Extraction of Viral Nucleic Acids

The genital tract virome was determined by shotgun sequencing of an aliquot (450 μL) from the vaginal lateral wall swab buffer solution. Virus-like particles (VLPs) in the vaginal swabs were resuspended by the addition of 1 mL SM buffer, 0.1 M NaCl, 50 mM Tris/HCl (pH 7.4), and 10 mM MgSO4, to the sample. This step was followed by filtering the sample through a 0.2 µm filter (Santorius, Göttingen, Germany) to separate VLPs from bacteria, human cells, and larger particles. Viral DNA was extracted from the resulting filtrate using the High Pure Viral Nucleic acid extraction kit (Roche Life Sciences, Indianapolis, IN, USA) following the manufacturer’s protocol. The extracted viral DNA was subjected to rolling circle amplification using the Illustra^TM^ TempliPhi kit (GE Healthcare, Chicago, IL, USA) and stored at −80 °C until library preparation.

### 2.6. Viral Metagenome Sequencing and Bioinformatics Analyses

Library preparation and quantitation were performed by the University of Washington (UW) Northwest Genomics Center (NWGC), and the resulting libraries were sequenced in multiplex mode on a NovaSeq6000 S4 flow cell, generating ~30 Gb per sample. The raw reads were trimmed using Trimmomatic 0.39 [23], and then de novo assembled using metaSPAdes 3.12.0 [24] with k = 33, 55, 77. *De novo* assembled contigs >500 nucleotides in length were identified via alignment against a viral protein RefSeq sequence database (downloaded from NCBI) using BLASTx [25]. Circular contigs were identified in the viral-like contigs by checking for terminal redundancy. All the open reading frames (ORF) were identified using ORFfinder at https://www.ncbi.nlm.nih.gov/orffinder/ (accessed 7 January 2022) and refined and annotated with data from PaVE [26]. The HPV genomes (*n* = 65) identified via viral metagenome sequencing are deposited in GenBank (OP970964-OP970967; OP971042-OP971102), and raw reads have been deposited in SRA under BioProject PRJNA881266.

### 2.7. Whole Community Metagenome Sequencing and Bioinformatics Analyses

We leveraged previously generated lateral vaginal wall shotgun metagenome sequencing data (i.e., from DNA extracted from swabs without VLP-enrichment, described in Happel et al. [27]) from participants of the same cohort to identify additional complete HPV genomes and report on these here. Briefly, the UW NWGC performed whole metagenome library preparation, quantitation and sequencing of genomic DNA using an Illumina NovaSeq 6000 S4 flow cell. The bioinformatic analysis has been described before [27], but for this analysis, circular genomes were identified based on terminal redundancy. Those with homology to previously described HPV genotypes (*n* = 45) have been deposited in GenBank (OP970997- OP971041). Raw metagenomic reads have been deposited in SRA under BioProject PRJNA767784.

### 2.8. Genome Analyses of HPVs and L1 Phylogeny

CenoteTaker2 [28], coupled with manually refined reference genomes from PaVE [26], was used to annotate the 110 HPV genomes. The classification of papillomavirus species is based on major capsid protein (L1) nucleotide sequence similarity [29]. The species demarcation threshold was 70%, with the papilloma-type threshold being 90%. The L1 sequences of classified papillomaviruses were downloaded from PaVE [26], including reference and non-reference sequences, and these sequences, together with the 110 genomes identified (65 genomes identified by viral metagenome sequencing and 45 genomes identified by whole community metagenome sequencing) constituted an L1 sequence dataset. The pairwise identities of L1 sequences were determined using SDT v1.2 [30]. Based on these pairwise identities of the L1 sequences, the HPVs identified in this study were assigned HPV types. Since the pairwise identities of the L1 sequences showed that all 110 HPVs are members of the *Alphapapillomavirus* or *Gammapapillomavirus* genus, we assembled a dataset of known L1 sequences of all HPV types in these two genera, as well as those in this study and a subset of those from the genera *Betapapillomavirus*, *Nupapillomavirus*, *Mupapillomavirus* and avian-infecting papillomaviruses. This dataset was translated and aligned using MAFFT v7.113 [31]. The alignment was then used to determine a best-fit amino acid substitution model (LG + I+G + F) using ProtTest 3 [32], and a maximum likelihood phylogenetic tree was inferred using PhyML 3 [33] with approximate likelihood-ratio test (aLRT) branch support. The maximum likelihood tree was rooted with the avian papillomavirus L1 sequences and visualized and annotated in iTOL v6 [34].

### 2.9. Statistical Analyses

All downstream analyses were conducted in Rstudio using R version 4.2.2 [35] unless otherwise specified. Study cohort characteristics were described using means, medians, standard deviations, interquartile ranges, and proportions, as appropriate. Differences in study population characteristics during the absence and presence of BV were tested using Fisher’s exact test for count measurements, paired Student’s t-test for differences in means (parametric data) and paired Wilcoxon signed-rank test for differences in medians (nonparametric data). Co-occurrence patterns of HPV genotypes were determined using the R package cooccur [36]. The Data Integration Analysis for Biomarker discovery using Latent Components (DIABLO) framework, as part of the mixOmics R Bioconductor package [37], was used to integrate the microbial and HPV data. Bacterial taxa that accounted for the highest degree of variance between women with and without HPV were selected via sparse partial least-squares discriminant analysis (sPLS-DA) using centered log ratio (CLR) transformed bacterial relative abundance and binary presence/absence of any HPV. A network including the selected variable was drawn using the network function and visualized using the igraph R package version 1.3.5 [38] and reformatted in Cytoscape [39]. Differences in median cervicovaginal cytokine concentrations between women with any HPV, multiple HPVs, any high-risk or any low-risk HPVs were compared to those without, and between women with a given HPV type versus those without that specific type using unpaired Mann-Whitney U test (nonparametic data). Mixed effect logistic regressions that included participant identification number (PID) as a random variable (since a subset of participants had samples collected at two time points) and adjusted for BV status (since BV influences cytokine concentrations) were run using the R package lme4 [40] to evaluate associations between HPV positivity and concentrations of cervicovaginal cytokines. Adjustment for multiple comparisons was done using a false discovery rate step-down procedure [41]. *p* values and 95% confidence intervals were used to assess statistical significance.

## 3. Results

### 3.1. The Eukaryotic Vaginal DNA Virome in South African Adolescents

The 33 South African adolescents and young women included in the viral metagenome analysis had a median age of 16 years (IQR 16–18 years) with a median age of sexual debut of 15 years (IQR 14–16 years). Most of the sampling was performed when participants were using injectables, oral contraceptive pills or Nuvaring^®^ as hormonal contraceptives (Table 1). The prevalence of HSV-2 seropositivity and BV was 30%, and about one-fifth had a bacterial STI or candidiasis. With regards to sexual risk behavior, most women only had one partner, half used a condom during their last sex act, and one-sixth were unsure whether their partner was monogamous. Previous pregnancies were rare (Table 1). About half of the women had a vaginal microbiota dominated by a diverse group of anaerobic bacteria, with the most abundant species being *Gardnerella vaginalis* followed by *Lachnocurva vaginae* (formerly BVAB1), *Megasphaera, L. iners, Prevotella* spp. (including *P. amnii*, *P. timonensis*, and *P. bivia*), *Fanyhessa vaginae*, *Sneathia*, and *Aerococcus christensenii* (community state type IV (CST-IV), Appendix A). The remainder had low-diversity communities dominated by *L. crispatus* (CST-I) or *L. iners* (CST-III). For a subset of eight participants that experienced a change in BV status, the vaginal viromes of longitudinal samples were sequenced from visits both with and without BV, 4 months apart. As expected, participants had a more diverse bacterial microbiota (CST-IV) when diagnosed with BV, and most vaginal microbiotas were dominated by *Lactobacillus* spp. (CST-I or -III) with a lower diversity in the absence of BV. A higher proportion of women were using oral contraceptive pills during episodes of BV, but no other evaluated demographic or biological variable differed by BV status (Table 1).

To identify eukaryote-infecting DNA viruses in the vaginal samples, we characterized the viral taxonomy of contigs longer than 500 nucleotides using BLASTx and the NCBI Refseq viral protein database. Sequences from 11 eukaryote-infecting virus families were detected (Figure 1). We identified ssDNA viruses that share sequence similarities to *Anelloviridae* and *Genomoviridae*; and dsDNA viruses that share sequence similarities to *Adenoviridae*, *Alloherpesviridae*, *Herpesviridae,* in the subfamilies *Alphaherpesvirinae*, *Betaherpesvirinae*, and *Gammaherpesvirinae*, *Marseilleviridae*, *Mimiviridae*, *Polyomaviridae*, *Poxviridae* in the subfamilies *Chordopoxvirinae* and *Entomopoxvirinae*. We also observed sequences with similarities to *Retroviridae,* primarily human endogenous retroviruses. Finally, we identified complete genomes in the viral families *Anelloviridae* (*n* = 3), *Genomoviridae* (*n* = 1), and *Polyomaviridae* (*n* = 2), which have been reported by Jimoh et al. [42].

### 3.2. HPV Infection Patterns in South African Adolescents

Among the eukaryote-infecting viruses, we identified multiple viruses in the *Papillomaviridae* (subfamily *Firstpapillomavirinae*) family, which are all HPVs. We assembled 65 full genomes from the viral metagenomics dataset and leveraged our total metagenome dataset from a subset of participants from the same cohort [27] to assemble an additional 45 HPV genomes. These 110 unique genomes represent 40 types and 12 species within two genera, i.e., *Alphapapillomavirus* (*n* = 104) and *Gammapapillomavirus* (*n* = 6) (Table 2; Figure 2).

The overall prevalence of any HPV in this group of women was 60.6% (20/33), with 54.5% (18/33) being infected with high-risk HPV types. Most of the women infected with any HPV were infected with multiple HPV types (80.0%; 16/20). The most frequent HPV types detected were HPV-35 (high-risk), found in six women, followed by HPV-16 and -53 (both high-risk) and HPV-81 and -90 (both low-risk). HPV-16, -53, -81, and -90 were each detected in four women. Interestingly, one woman had HPV-214, which was only recently described in a penile swab from a study conducted in Cape Town, South Africa [43]. HPV types targeted by the bivalent HPV vaccine (HPV-16/-18) were detected in 15.2% (5/33) of women, while those targeted by the quadrivalent vaccine (HPV-6/-11/-16/-18) were detected in 21.2% (7/33) of women; and those targeted by the nonavalent vaccine (HPV-6/-11/-16/-18/-31/-33/-45/-52/-58) were detected in 33.3% (11/33) of women. HPV-11, which is one of the types targeted by the quadrivalent vaccine, was not detected, while all other HPV types targeted in the bivalent, quadrivalent and nonavalent vaccines were found. The most common HPV type identified (HPV-35) is, however, not targeted by any of the commercially available vaccines.

We next evaluated HPV co-infection patterns using a model that employs combinatorics to determine the probability that the observed frequency of HPV genotype co-occurrence is significantly greater than expected (positive association), significantly less than expected (negative association), or approximately equal to expected (random association) (Figure 3a). Of the 40 HPV genotypes identified in this study, 35 showed positive associations with one or more HPV genotypes. No associations were found between HPV-62, -66, -67, -215, -mSD2 and any other HPV genotype. There was a significant positive co-occurrence between the presence of HPV-16 and twelve other HPV genotypes, including high-risk HPV-18, -33, -53, -56, and -68, and low-risk HPV-30, -40, -43, -44, -54, -61 and -69 (all *p* < 0.05) (Figure 3a). In agreement, samples where HPV-16 was present (*n* = 6) had significantly more other HPV types (median 12, interquartile range (IQR) 5–12) compared to samples where HPV-16 was not present (*n* = 35; median 1, IQR 0–2; *p* < 0.0001). Low-risk HPV-61 and high-risk HPV-69 co-occurred in all samples in which the given genotypes were detected (*p* = 0.0008). This was also true for the co-occurrence of HPV-39 and -68 (both high-risk, *p* = 0.0003) and HPV-40 and -53 (both low-risk, *p* = 0.0003). These data suggest that HPV genotypes frequently co-occur and that the co-occurrence between specific HPV genotypes might be more common than for others.

### 3.3. HPV and the Vaginal Microbiota

Given the previously reported relationship between HPV and the vaginal microbiota [16], we evaluated associations between the presence of HPV and the vaginal microbiota in cross-sectional analyses that included 19 samples collected when women had BV and 22 samples collected during the absence of BV, using an untargeted viral metagenome sequencing approach. Cervicovaginal samples from women with BV had a significantly higher number of HPV genotypes (median 2, IQR 0–6) compared to those without BV (median 0, IQR 0–3, *p* = 0.031). The proportion of women with high-risk HPV types was not statistically different between women with BV (12/19, 63.2%) compared to those without BV (10/22, 45.5% *p*= 0.350). Twelve of the 39 HPV genotypes included in this analysis (30.7%) only occurred in women with BV, including high-risk types HPV-35 (present in 5/19 (26.3%) of women with BV), HPV-26 (*n* = 1) and HPV-31 (*n* = 1), and low-risk genotypes HPV-40 (*n* = 3), HPV-6, -42, -74, -81, -91 (all *n* = 2), HPV-84, -108 and -214 (all *n* = 1) with unknown risk association (Figure 3b).

We performed sPLSDA to identify bacterial taxa that accounted for the highest degree of variance between women with and without any HPV (Figure 4a). A network analysis that included the statistically significant bacterial taxa selected by sPLSDA showed that the presence of HPV correlated positively with the presence of *Coriobacteriaceae, Prevotella* and *Dialister* spp., in addition to other BV-associated species, including *Megasphera*, *Gardnerella vaginalis, Fannyhessea vaginae and Aerococcus christensenii* (Figure 4b)*.* In contrast, the absence of HPV infection correlated positively with the presence of *L. crispatus,* and *Corynebacteria (C. tuberculostearicum* and *C. amycolatum)*. There were two main clusters of bacteria that best distinguished HPV status: one dominated by *L. crispatus* (with a lower abundance in women infected with HPV) and one comprised of *Coriobacteriaceae*, *Prevotella* spp. (*P. buccalis* and *P. timonensis*), *Parvimonas micra* and *Dialister micraerophilus* (with a higher abundance in women infected with HPV). Most of the latter clusters were highly abundant in CST-IV in this cohort (Appendix A) and have previously been associated with BV and genital inflammation in South African women [44,45]. In agreement, most samples assigned to this BV-associated bacteria cluster (16/20, 80%) were from women who had HPV, while fewer women assigned to the *L. crispatus* cluster had HPV (9/21; 43%; *p* = 0.0247) (Figure 4c). These data suggest that vaginal microbiota and HPV infection might be associated in this cohort of South African adolescents.

We sought to investigate this relationship in more detail in a subset of eight participants who experienced a change in BV status longitudinally over 4 months (Table 1, Figure 5). This analysis only included samples with a read coverage of >75% of the genome as a cut-off for positive detection of a given HPV type. Using these criteria, four women had more HPV variants present during episodes of BV compared to episodes with no BV, three had fewer, and one participant did not have any HPV in the presence or absence of BV. Women had a median of three HPV variants during a BV episode (IQR 1–8) and a median of two variants when BV was absent (IQR 0–4; *p* = 0.638). The same observations were made on genotype (median 2 (IQR 0–4) vs. 3 (IQR 1–4), *p* = 0.678) and species (median 2 (IQR 0–3) vs. 2 (IQR 1–5), *p* = 0.674) level. This suggests that while there were associations between the presence of HPV and some BV-associated taxa, and for some women episodes of BV coincide with the acquisition of new HPV genotypes, this pattern is not universal.

### 3.4. HPV and Cervicovaginal Cytokine Concentrations

Previously, in a study of South African women at high risk of HIV acquisition, infection with any HPV type was associated with increases in several chemokines, growth/hematopoietic factors, pro-inflammatory, adaptive and regulatory cytokines, including those found to be associated with HIV risk [46]. Similar observations were made in other settings [47,48].

Here, we compared cervicovaginal cytokine levels from women infected with a given HPV genotype to those without that specific genotype in univariate analyses (Figure 6a). Infection with some HPV types (i.e., high-risk HPV-35, -39, and -68) was associated with higher cervicovaginal cytokine concentrations, particularly of IL-17A, IL-17F, TNF-α, IL-25 and IFN-γ. In contrast, infection with HPV genotypes of the species *Alphapapillomovirus* 6 (i.e., HPV-30) was generally associated with lower cytokines concentrations. In univariate analyses, infection with HPV-35 was associated with a significantly higher concentration of six of the thirteen cytokines measured, including IL-1β (median 131.65 pg/mL (IQR 109.08–214.31) vs. 7.15 pg/mL (0.61–59.33), *p* = 0.0009), TNF-α (median 20.12 pg/mL (IQR 6.21–22.94) vs. 0.68 pg/mL (0.16–1.93), *p* = 0.0009), IL-33 (median 24.24 pg/mL (IQR 11.70–25.02) vs. 5.85 pg/mL (IQR 1.68–9.86), *p* = 0.012), IL-17A (median 3.80 pg/mL (IQR 2.74–4.50) vs. 1.24 pg/mL (IQR 0.82–2.82), *p* = 0.037), IL-6 (median 4.24 pg/mL (IQR 2.96–17.26) vs. 0.95 (IQR 0.47–2.73), *p* = 0.043) and IL-17F (median 22.23 pg/mL (IQR 11.33–23.08) vs. 3.40 pg/mL (IQR 0.34–8.89), *p* = 0.048), with IL-1β and TNF-α remaining significant after adjustment for multiple comparisons (adjust. *p* = 0.0044 and adjust. *p* = 0.0054, respectively). As BV has been shown to increase genital inflammation, we adjusted for BV status in mixed effect logistic regressions that included PID as a random variable. In these models, only TNF-α showed a significant association with the presence of HPV-35 (β = 2.53; 95% CI 1.30–1.95, *p* = 0.048).

To further dissect the relationship between HPV and genital cytokines, we compared cervicovaginal cytokine levels from women according to HPV status, including infections with any HPV, multiple HPVs, high-risk, or low-risk HPVs versus those without any HPV infection (Figure 6b). We found no significant differences in cytokine concentrations in univariate analyses between women with any, multiple or any high-risk or low-risk HPVs compared to those without. We next used mixed effect logistic regressions that included PID as a random variable and adjusted for BV status to evaluate associations between HPV positivity and levels of cervicovaginal cytokines (Table 3). This analysis confirmed that having any HPV, multiple HPVs, any high-risk or any low-risk HPVs was not associated with significantly higher concentrations in any of the cervicovaginal cytokines measured, while having BV was associated with significantly higher IL-1β (β = 1.00; 95% CI 0.30 -1.69; *p* = 0.006), IL-17F (β = 0.82; 95% CI 0.01–1.64; *p* = 0.048), IL-25 (β = 0.68; 95% CI 0.03–1.33; *p* = 0.041), IL-33 (β = 1.34; 95% CI 0.12–2.57; *p* = 0.032) and TNF-α (β = 1.60; 95% CI 0.52–2.68; *p* = 0.004). This suggests that the presence of BV has a greater impact on cytokine levels in the lower FGT than the presence of HPV in this group of women.

We controlled for BV and included PID as a random variable, as some participants had repeated measurements. *p*-values after adjustment for multiple comparisons using a false discovery rate step-down procedure [41] are shown.

## 4. Discussion

Our exploratory study describes the eukaryote-infecting DNA virome in young women from Sub-Saharan Africa, with an emphasis on HPV genomes. Using a metagenome sequencing approach, we assembled 110 complete HPV genomes across 40 HPV genotypes. We have made these publicly available to expand the current databases.

This untargeted approach for HPV typing allowed us to capture a broader snapshot of the FGT HPV landscape, as we were not limited to HPV types included in commercial arrays. While most HPV types detected are included in currently available commercial assays like the HPV Direct Flow CHIP (Master Diagnóstica, Granada, Spain), more recently described HPV genotypes like HPV-108, -214, -215 or -mSD2 would not have been detected using commercial arrays. The risks associated with infections with these HPV genotypes are still unknown and will be difficult to assess if only commercial arrays are used to conduct HPV typing.

HPV-35 was the most common HPV type detected, with a prevalence of 15% in this small cohort of asymptomatic, young South African women. Studies in Sub-Saharan African populations have reported an HPV-35 prevalence of up to 40% among women with cervical intraepithelial neoplasia (CIN) [49]. HPV-35 is also detected in approximately 10% of cervical cancer cases in African women, while it is detected in 2% of cervical cancer cases worldwide [50]. Despite its high prevalence and risk associated with cancer development in African women, HPV-35 is not targeted by any of the currently available HPV vaccines. HPV co-infections were common, especially with high-risk HPV genotypes. Whether this relates to behavior or biological mechanisms remains unclear. Our cohort included young women, and others have found that co-infections are more common amongst younger women than their older counterparts [51,52], suggesting that the higher frequency of sexual activity of younger women, or the higher number of partners, may lead to the sexual transmission of multiple HPV genotypes. HPV typing and investigation of co-infection patterns on a larger scale across more demographic and age groups would help inform the HPV vaccination campaigns for Sub-Saharan African women. Given the high prevalence of HPV in this age group, alongside high HIV risk [10], and recent observations that infection with any high-risk or low-risk HPV is associated with increased risk of HIV acquisition in Sub-Saharan women [9], these findings not only provide an argument for increased HPV vaccine coverage to prevent HPV-associated cancers but also to potentially reduce HIV incidence in this population. Further, while the current South African guidelines recommend that all women should have a Pap smear at least every 10 years, starting at age 30, this might suggest including younger women in HPV screening programs, in addition to raising awareness around this topic [53].

A meta-analysis has suggested a causal link between vaginal dysbiosis and HPV risk [16]. There was limited evidence for associations between the presence of BV by Nugent scoring and HPV infections in our study, but the sample size is small and limits our power to detect differences. However, in a more detailed analysis of the vaginal microbiota composition, we found that women with HPV had less *Lactobacillus* spp. and a higher abundance of *Prevotella buccalis* and *P. timonensis*, *Parvimonas micra* and *Dialister micraerophilus,* which were highly abundant in women with CST-IV in this cohort. This supports findings from a recent study that has described that HPV infection alters the vaginal microbiome through the down-regulation of host mucosal innate peptides used by *Lactobacillus* spp. as amino acid sources [54]. Larger longitudinal studies are needed to infer causality and describe the directionality in the interplay between HPV and BV. Our findings raise the question of whether the shifting of the vaginal microbiota towards a *Lactobacillus*-dominant state, e.g., by using live biotherapeutic products, could be leveraged as a treatment strategy for women with HPV. While larger studies in diverse populations are warranted, previous studies have suggested that the administration of *Lactobacillus* probiotics may favor HPV clearance [55,56,57].

Following the adjusting for BV status, we found that having any HPV type or a high-risk HPV type was not associated with cytokine levels in this small cohort of women. Only infection with HPV-35 was associated with significantly higher concentrations of TNF-α after adjusting for BV status. In contrast, having BV was associated with higher levels of many cytokines that overlap substantially with cytokines associated with HIV risk. While HPV infection has been found to increase the risk of HIV acquisition in Sub-Sharan African women [9], and an increase in genital inflammation, independent of the cause, has been associated with an increased risk of HIV acquisition [11], these data suggest that the increased risk of HIV acquisition amongst women with HPV is not facilitated via an increase in genital inflammation.

The sample size of this exploratory study was small, thereby limiting our ability to make strong conclusions regarding associations between vaginal microbiota, cytokines, and the presence of HPV genotypes. Nonetheless, we analyzed longitudinal, matched data for a subset of participants, which strengthened our approach. While we focused on DNA viruses here, we plan to include RNA viruses in subsequent studies and to assess bacteriophages. The effect of other clinical parameters on the vaginal microbiota, such as diet or vaginal insertion practices, should be considered in future studies to be carried out in different populations. Strengths of this study included using a viral metagenomic sequencing approach to capture the HPV landscape in a more unbiased way and to identify full HPV genomes from Sub-Saharan Africa, which we have made publicly available.

## 5. Conclusions

This viral metagenome sequencing study has identified 110 unique HPV genomes from healthy, asymptomatic South African women. Amongst the most common high-risk HPV types were genotypes that are not currently targeted by available vaccines. Describing the HPV landscape helps inform HPV vaccination strategies and should be conducted on a larger scale in Sub-Saharan African women.

## Figures and Tables

**Figure 1 viruses-15-00758-f001:**
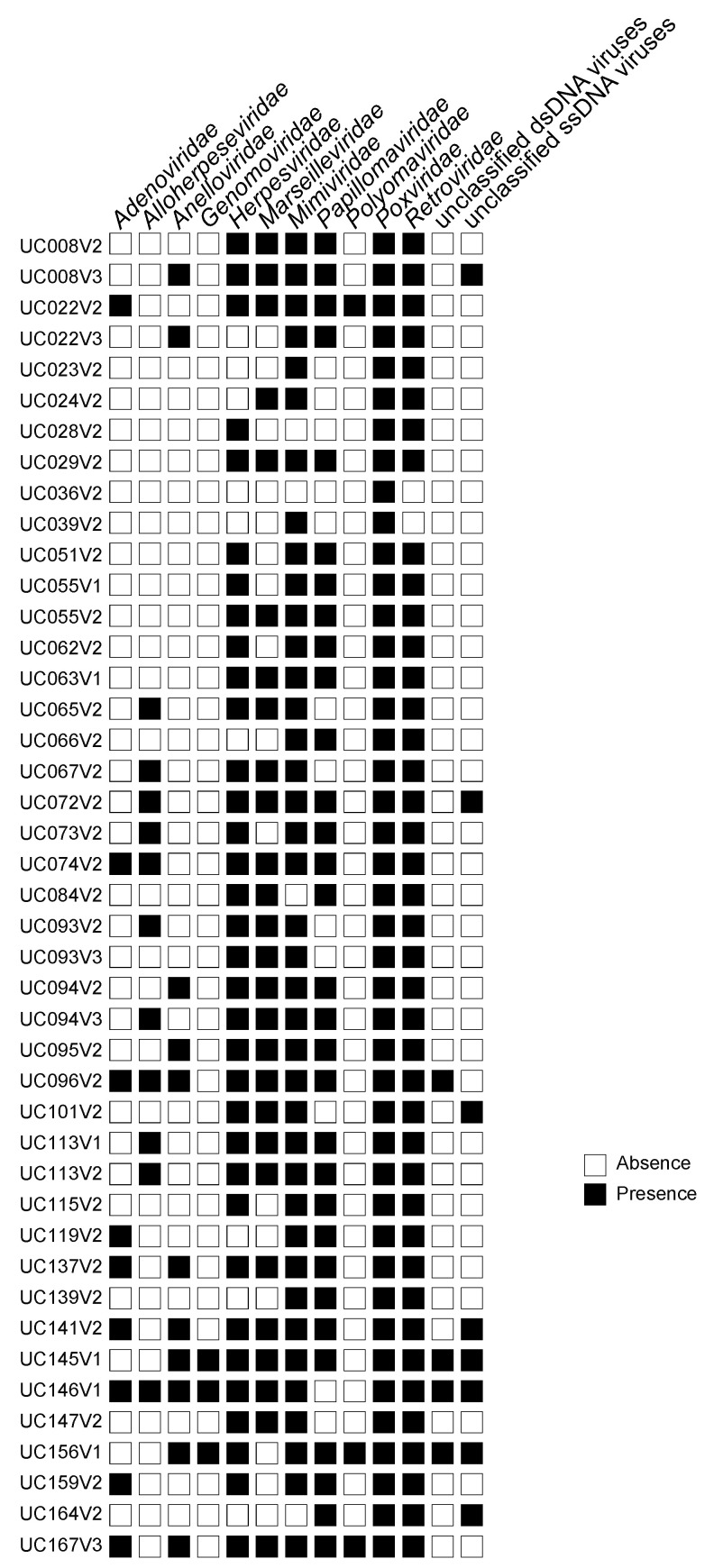
Eukaryotic DNA viruses in South African adolescents. Presence (black square) and absence (white square) of eukaryote-infecting families of DNA viruses identified by viral metagenome sequencing after rolling-circle-amplification of viral-nucleic acids extracted from vaginal samples from 33 South African women based on BLASTx analysis of contigs >500 bp against the NCBI viral Refseq database. PID and study visit at which samples were collected (V1 = baseline, V2 = 4 months, V3 = 8 months) are shown. This analysis includes samples from the 33 women included in the viral metagenome sequencing project.

**Figure 2 viruses-15-00758-f002:**
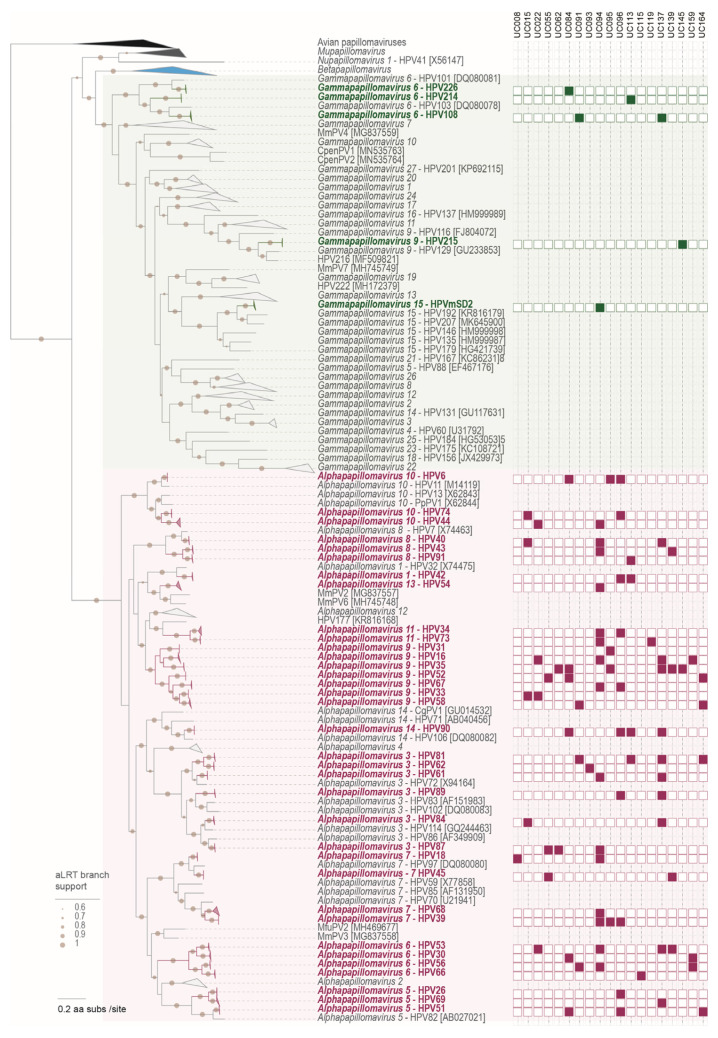
Maximum likelihood phylogeny of L1 amino acid sequences identified in this study. HPV species and types identified in 19 individuals are shown in bold font, and the colored boxes represent the presence of HPVs in that individual. Branches with >0.6 approximate likelihood-ratio test (aLRT) branch support are shown. This analysis includes all complete HPV genomes identified in the viral and whole community metagenome sequencing projects.

**Figure 3 viruses-15-00758-f003:**
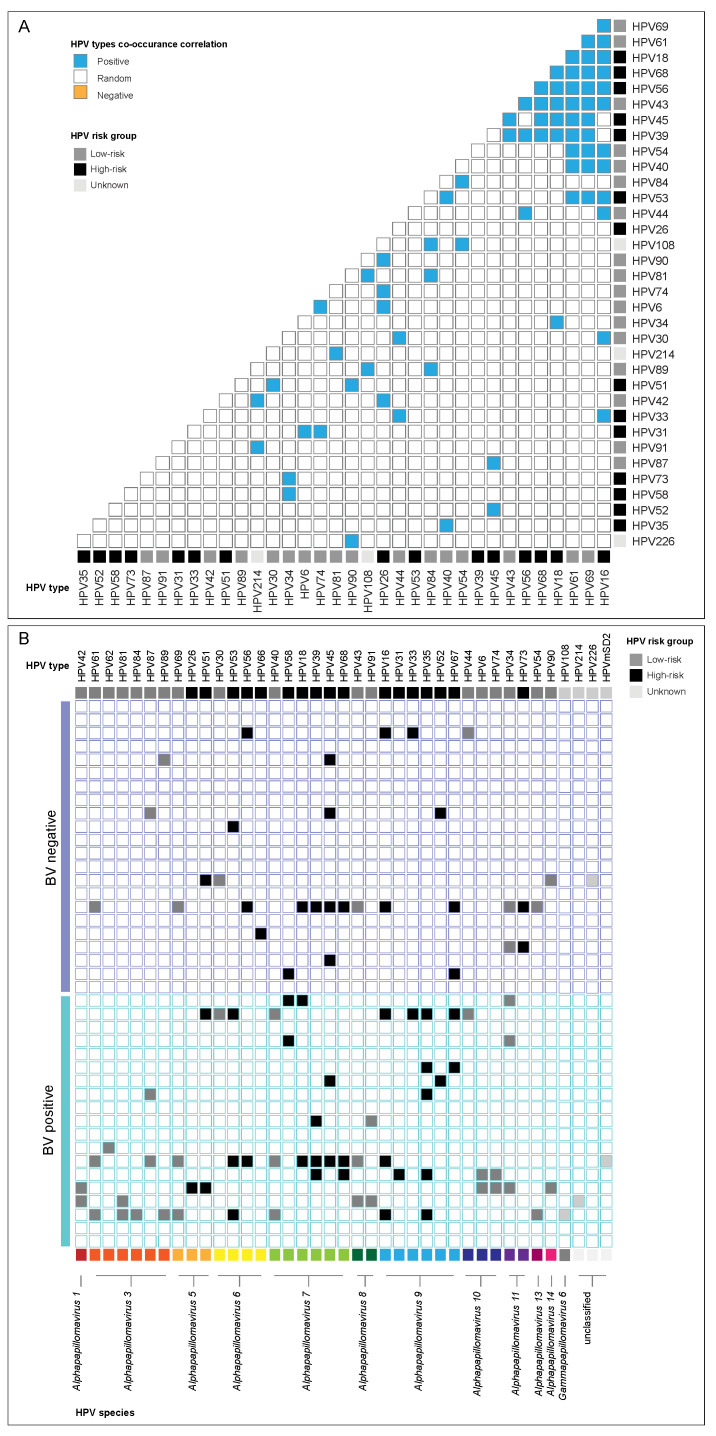
Occurrence of HPV genotypes. (**A**) Heat map showing the positive HPV genotype associations determined by a probabilistic co-occurrence model [36]. Genotype names are positioned to indicate the columns and rows that represent their pairwise relationships with other species. Blue indicates that the co-occurrence of HPV genotypes is significantly large and greater than expected (statistically significant positive association). This analysis included the HPV genotypes identified in the viral metagenome and whole community sequencing projects. The plot only includes the 35 HPV genotypes that had significant associations with one or more other HPV genotypes. (**B**) The occurrence of the 39 HPV genotypes identified in the 33 participants collected when women had BV (light blue, *n* = 19) and when BV was absent (dark blue, *n* = 22) is shown in a cross-sectional comparison. HPV genotypes are annotated by risk classification and species. This analysis includes samples from the 33 women in the viral metagenome sequencing project.

**Figure 4 viruses-15-00758-f004:**
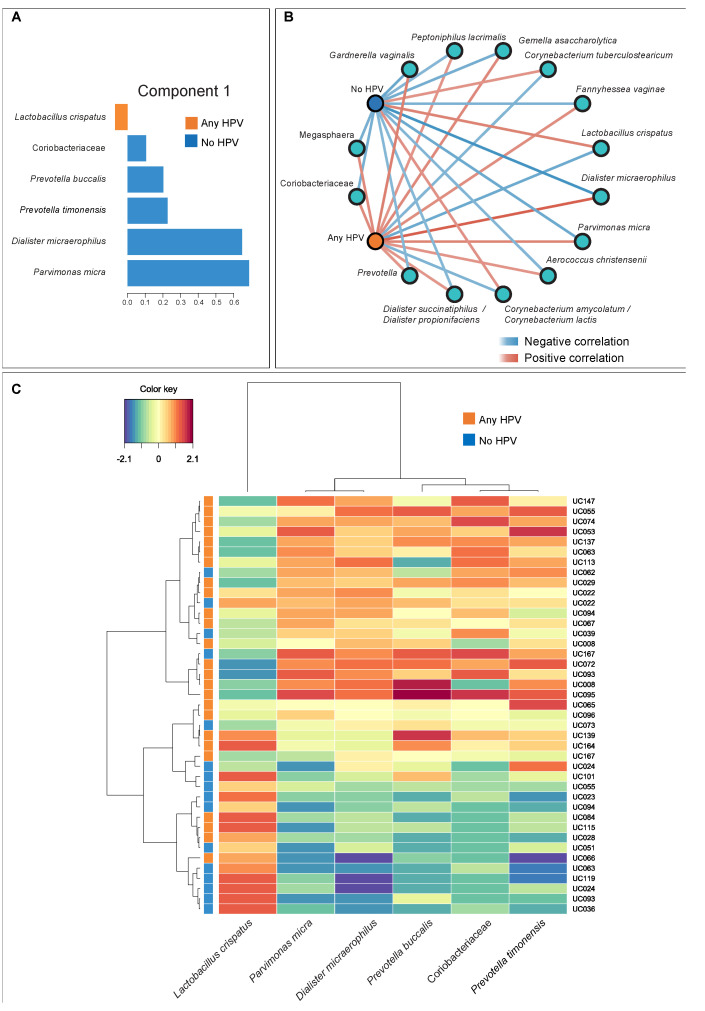
Distinguishing women with HPV versus without HPV based on vaginal microbiota composition. (**A**) Loadings plot showing the contribution of the different bacterial taxa selected by sparse PLS-DA (sPLSDA) that explain the highest variance in the comparison of samples from women with any HPV (orange) and from women without any HPV (blue). (**B**) A network analysis of the bacterial taxa selected by sPLSDA. Positive correlations are shown in red, while negative correlations are shown in blue. (**C**) Unsupervised clustering of the selected bacterial taxa, annotated by HPV status with orange indicating the presence of any HPV and blue indicating the absence of HPV in a sample. This analysis includes samples from the 33 women included in the viral metagenome sequencing project.

**Figure 5 viruses-15-00758-f005:**
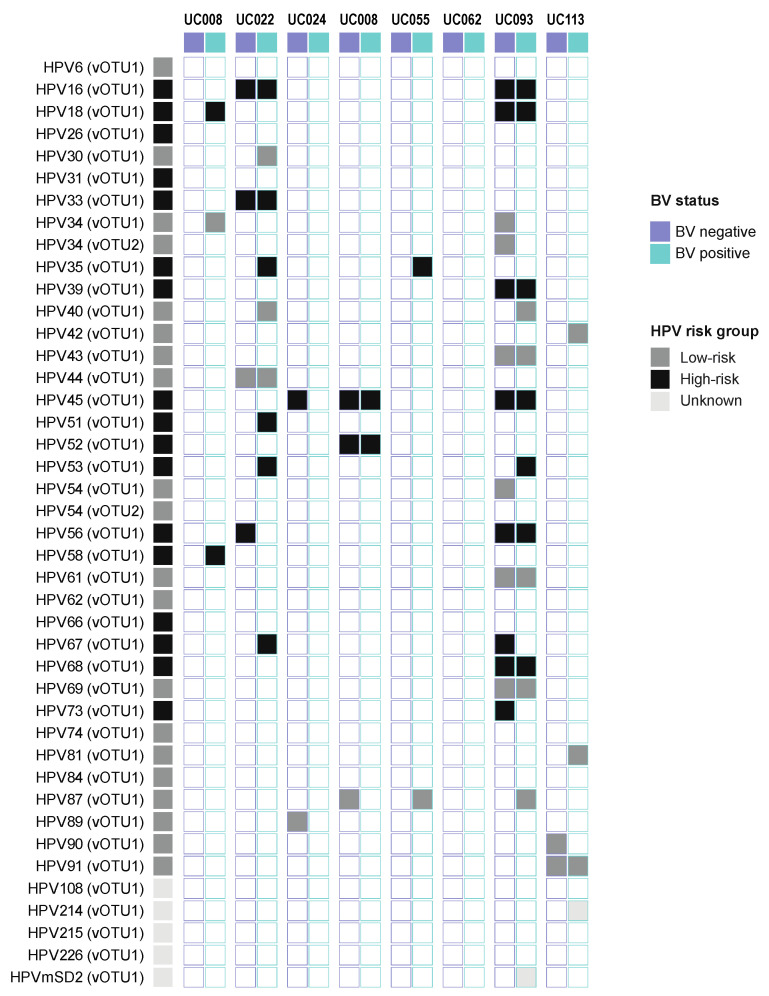
Evaluating presence of HPV variants in longitudinal samples from participants who experienced a change in BV status. From eight participants included in the viral metagenome sequencing project longitudinal samples were collected 4 months apart, and HPV genotype presence was evaluated in the absence (purple) and presence (blue) of BV. A variant was defined based on a 5% difference (virus operational taxonomic unit, vOTU), and the analysis was limited to vOTUs that had >75% genome coverage.

**Figure 6 viruses-15-00758-f006:**
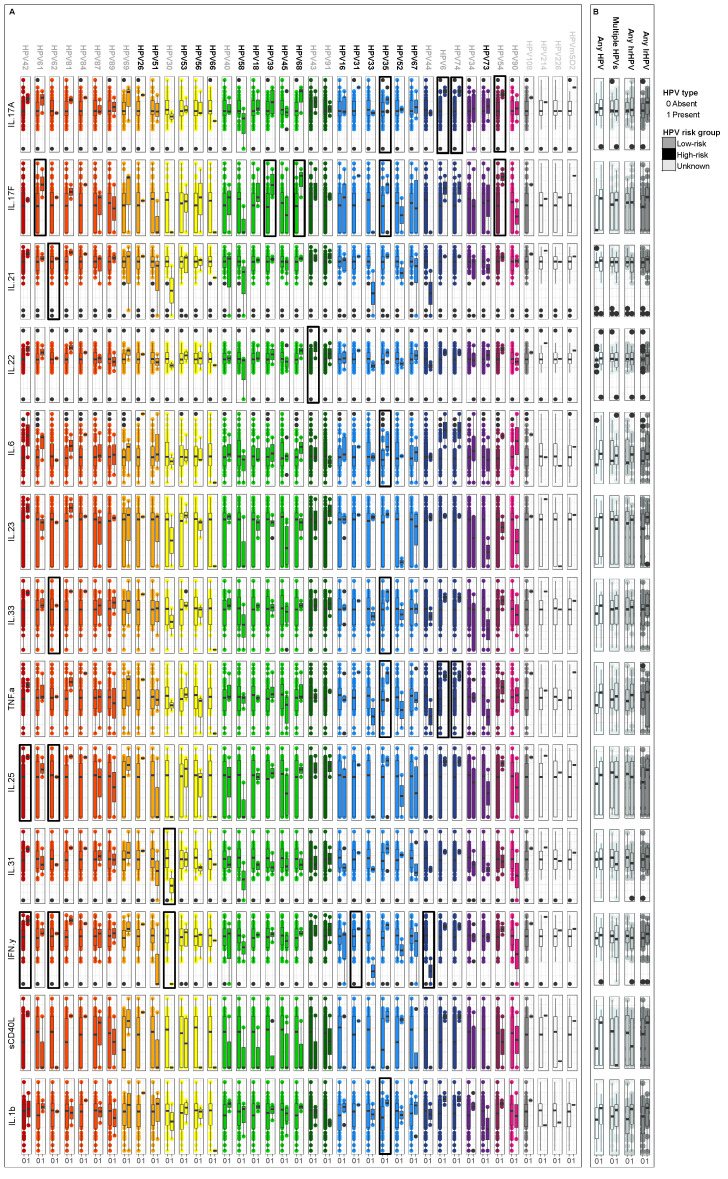
Associations between cervicovaginal cytokines and presence of HPV genotypes. (**A**) In a univariate analysis, the measured concentrations of cytokines were compared in samples of women with a given HPV genotype (labelled 1) to that of samples from women without the given HPV genotype (labelled 0). Significant differences are highlighted using a black box. (**B**) The same comparisons were made for samples from women with any HPV genotype present versus those without, and those with multiple HPV genotypes, any high- (hr) or any low-risk (lr) HPV genotypes. This analysis includes samples from the 33 women included in the viral metagenome sequencing project.

**Table 1 viruses-15-00758-t001:** Characteristics of participants included in the viral metagenome analysis.

	All (*n* = 33)	BV − (*n* = 8) ^#^	BV + (*n* = 8) ^#^	*p*-Value ^#^
Age at screening, years [median (IQR)]	16 (16–18)	17 (15–18)	-	-
BMI [median (IQR)]	24.6 (21.8–29.3)	23.6 (22.5–26.2)	24.5 (23.1–30.3)	0.482
Contraceptive use [n(%)]		data	data	data
None	1 (3.0)	1 (12.5)	0 (0)	>0.999
Injectable	11 (33.3)	2 (25.0)	2 (25.0)	>0.999
COC	7 (21.2)	0 (0)	5 (62.5)	0.025
Nuvaring^®^	14 (42.4)	5 (62.5)	1 (12.5)	0.119
Vaginal pH [median (IQR)]	4.6 (4.2–5.2)	4.7 (4.2–5.0)	4.7 (4.4–4.9)	0.873
Sexually-transmitted infections [n(%)]				
HSV-2 seroprevalence ^a^	10 (30.3)	3 (37.5)	3 (37.5)	>0.999
*N. gonorrhea*	0 (0)	1 (12.5)	0 (0)	>0.999
*C. trachomatis ^b^*	4 (12.1)	2 (25.0)	1 (12.5)	>0.999
*M. genitalium ^b^*	1 (3.0)			
Any bacterial STI	5 (15.1)	3 (37.5)	1 (12.5)	0.569
Candidiasis [n(%)]	6 (18.2)	4 (50.0)	1 (12.5)	0.282
Nugent-BV [n(%)]	12 (36.4)	0 (0)	8 (100)	0.0002
CST distribution [n(%)]				
CST-I	9 (27.3)	1 (12.5)	0 (0)	>0.999
CST-III	9 (27.3)	6 (75.0)	0 (0)	0.007
CST-IV	15 (45.5)	1 (12.5)	8 (100)	0.0004
Shannon diversity [mean (sd)]	1.5 (0.5–1.8)	0.6 (0.4–1.5)	2.1 (1.6–2.2)	0.032
Sexual Risk Behavior				
Age sexual debut, years [median (IQR)]	15 (14–16)	15 (14–16)	-	-
Number of sexual partners [mean (SD)] ^c^	1 (0.3)	1 (0)	1 (0)	>0.999
Participant unsure if partner has multiple partners [n(%)] ^d^	4 (13.3)	2 (25.0)	3 (37.5)	>0.999
Condom use during last sex act [n(%)] ^d^	15 (50.0)	5 (62.5)	3 (37.5)	0.619
Number vaginal sex acts per week [mean (SD)] ^d^	0.9 (0.3)	2 (1.2)	2 (0.8)	>0.999
Previously pregnant [n(%)] ^f^	2 (6.25)	1 (12.5)	-	-

BMI; body mass index, BV; bacterial vaginosis, COC; combined oral contraceptives, CST; community state type, HSV; herpes simplex virus, IQR; interquartile range, SD; standard deviation, STI; sexually transmitted infections. ^#^
*p*-value comparing characteristics of 8 participants in the absence versus presence of BV using Fisher’s exact test, paired Wilcoxon signed-rank test or paired Student’s t-test as appropriate. ^a^ Seroprevalence was assessed in blood by ELISA; ^b^ Active infection was assessed in genital swabs by PCR assays, ^c^ data missing for *n* = 7; ^d^ data missing for *n* = 3; ^f^ data missing for *n* = 1.

**Table 2 viruses-15-00758-t002:** List of identified complete HPV genomes determined in this study.

Virus Name	Genus	Species	Type	GenBank Accession
HPV6_UC095_LW_V2	*Alphapapillomavirus*	*Alphapapillomavirus 10*	human papillomavirus 6	OP971086
HPV6_UC096_LW_V2	*Alphapapillomavirus*	*Alphapapillomavirus 10*	human papillomavirus 6	OP971088
HPV6_UC084_LW_V1	*Alphapapillomavirus*	*Alphapapillomavirus 10*	human papillomavirus 6	OP971006
HPV6_UC084_LW_V2	*Alphapapillomavirus*	*Alphapapillomavirus 10*	human papillomavirus 6	OP971011
HPV6_UC096_LW_V3	*Alphapapillomavirus*	*Alphapapillomavirus 10*	human papillomavirus 6	OP971026
HPV16_UC022_LW_V2	*Alphapapillomavirus*	*Alphapapillomavirus 9*	human papillomavirus 16	OP971070
HPV16_UC094_LW_V2	*Alphapapillomavirus*	*Alphapapillomavirus 9*	human papillomavirus 16	OP971046
HPV16_UC094_LW_V3	*Alphapapillomavirus*	*Alphapapillomavirus 9*	human papillomavirus 16	OP971080
HPV16_UC137_LW_V2	*Alphapapillomavirus*	*Alphapapillomavirus 9*	human papillomavirus 16	OP971062
HPV16_UC159_LW_V2	*Alphapapillomavirus*	*Alphapapillomavirus 9*	human papillomavirus 16	OP971067
HPV18_UC008_LW_V3	*Alphapapillomavirus*	*Alphapapillomavirus 7*	human papillomavirus 18	OP971042
HPV18_UC094_LW_V2	*Alphapapillomavirus*	*Alphapapillomavirus 7*	human papillomavirus 18	OP971048
HPV18_UC094_LW_V3	*Alphapapillomavirus*	*Alphapapillomavirus 7*	human papillomavirus 18	OP971083
HPV26_UC096_LW_V2	*Alphapapillomavirus*	*Alphapapillomavirus 5*	human papillomavirus 26	OP971091
HPV30_UC159_LW_V2	*Alphapapillomavirus*	*Alphapapillomavirus 6*	human papillomavirus 30	OP971102
HPV30_UC084_LW_V1	*Alphapapillomavirus*	*Alphapapillomavirus 6*	human papillomavirus 30	OP971007
HPV30_UC084_LW_V2	*Alphapapillomavirus*	*Alphapapillomavirus 6*	human papillomavirus 30	OP971013
HPV31_UC095_LW_V2	*Alphapapillomavirus*	*Alphapapillomavirus 9*	human papillomavirus 31	OP970964
HPV33_UC022_LW_V2	*Alphapapillomavirus*	*Alphapapillomavirus 9*	human papillomavirus 33	OP971069
HPV33_UC015_LW_V1	*Alphapapillomavirus*	*Alphapapillomavirus 9*	human papillomavirus 33	OP970997
HPV34_UC094_LW_V2	*Alphapapillomavirus*	*Alphapapillomavirus 11*	human papillomavirus 34	OP971076
HPV34_UC096_LW_V2	*Alphapapillomavirus*	*Alphapapillomavirus 11*	human papillomavirus 34	OP971093
HPV34_UC084_LW_V1	*Alphapapillomavirus*	*Alphapapillomavirus 11*	human papillomavirus 34	OP971009
HPV35_UC062_LW_V2	*Alphapapillomavirus*	*Alphapapillomavirus 9*	human papillomavirus 35	OP971043
HPV35_UC095_LW_V2	*Alphapapillomavirus*	*Alphapapillomavirus 9*	human papillomavirus 35	OP971054
HPV35_UC137_LW_V2	*Alphapapillomavirus*	*Alphapapillomavirus 9*	human papillomavirus 35	OP971063
HPV35_UC145_LW_V1	*Alphapapillomavirus*	*Alphapapillomavirus 9*	human papillomavirus 35	OP971100
HPV35_UC139_LW_V2	*Alphapapillomavirus*	*Alphapapillomavirus 9*	human papillomavirus 35	OP971032
HPV35_UC139_LW_V3	*Alphapapillomavirus*	*Alphapapillomavirus 9*	human papillomavirus 35	OP971034
HPV39_UC094_LW_V2	*Alphapapillomavirus*	*Alphapapillomavirus 7*	human papillomavirus 39	OP971074
HPV39_UC094_LW_V3	*Alphapapillomavirus*	*Alphapapillomavirus 7*	human papillomavirus 39	OP971084
HPV39_UC095_LW_V2	*Alphapapillomavirus*	*Alphapapillomavirus 7*	human papillomavirus 39	OP971087
HPV39_UC096_LW_V1	*Alphapapillomavirus*	*Alphapapillomavirus 7*	human papillomavirus 39	OP971024
HPV40_UC094_LW_V3	*Alphapapillomavirus*	*Alphapapillomavirus 7*	human papillomavirus 40	OP971079
HPV40_UC137_LW_V2	*Alphapapillomavirus*	*Alphapapillomavirus 7*	human papillomavirus 40	OP971061
HPV40_UC015_LW_V3	*Alphapapillomavirus*	*Alphapapillomavirus 7*	human papillomavirus 40	OP970999
HPV42_UC096_LW_V2	*Alphapapillomavirus*	*Alphapapillomavirus 1*	human papillomavirus 42	OP971089
HPV42_UC113_LW_V2	*Alphapapillomavirus*	*Alphapapillomavirus 1*	human papillomavirus 42	OP971098
HPV42_UC096_LW_V3	*Alphapapillomavirus*	*Alphapapillomavirus 1*	human papillomavirus 42	OP971027
HPV43_UC094_LW_V2	*Alphapapillomavirus*	*Alphapapillomavirus 8*	human papillomavirus 43	OP971044
HPV43_UC094_LW_V3	*Alphapapillomavirus*	*Alphapapillomavirus 8*	human papillomavirus 43	OP971078
HPV43_UC139_LW_V1	*Alphapapillomavirus*	*Alphapapillomavirus 8*	human papillomavirus 43	OP971030
HPV44_UC022_LW_V2	*Alphapapillomavirus*	*Alphapapillomavirus 10*	human papillomavirus 44	OP971071
HPV45_UC094_LW_V2	*Alphapapillomavirus*	*Alphapapillomavirus 7*	human papillomavirus 45	OP971072
HPV45_UC094_LW_V3	*Alphapapillomavirus*	*Alphapapillomavirus 7*	human papillomavirus 45	OP971081
HPV45_UC055_LW_V2	*Alphapapillomavirus*	*Alphapapillomavirus 7*	human papillomavirus 45	OP971003
HPV45_UC139_LW_V2	*Alphapapillomavirus*	*Alphapapillomavirus 7*	human papillomavirus 45	OP971033
HPV51_UC096_LW_V2	*Alphapapillomavirus*	*Alphapapillomavirus 5*	human papillomavirus 51	OP971092
HPV51_UC084_LW_V1	*Alphapapillomavirus*	*Alphapapillomavirus 5*	human papillomavirus 51	OP971008
HPV51_UC084_LW_V2	*Alphapapillomavirus*	*Alphapapillomavirus 5*	human papillomavirus 51	OP971014
HPV51_UC164_LW_V1	*Alphapapillomavirus*	*Alphapapillomavirus 5*	human papillomavirus 51	OP971038
HPV52_UC055_LW_V1	*Alphapapillomavirus*	*Alphapapillomavirus 9*	human papillomavirus 52	OP971002
HPV52_UC084_LW_V2	*Alphapapillomavirus*	*Alphapapillomavirus 9*	human papillomavirus 52	OP971012
HPV52_UC164_LW_V1	*Alphapapillomavirus*	*Alphapapillomavirus 9*	human papillomavirus 52	OP971036
HPV52_UC164_LW_V2	*Alphapapillomavirus*	*Alphapapillomavirus 9*	human papillomavirus 52	OP971039
HPV52_UC164_LW_V3	*Alphapapillomavirus*	*Alphapapillomavirus 9*	human papillomavirus 52	OP971041
HPV53_UC022_LW_V3	*Alphapapillomavirus*	*Alphapapillomavirus 6*	human papillomavirus 53	OP970965
HPV53_UC094_LW_V3	*Alphapapillomavirus*	*Alphapapillomavirus 6*	human papillomavirus 53	OP971053
HPV53_UC137_LW_V2	*Alphapapillomavirus*	*Alphapapillomavirus 6*	human papillomavirus 53	OP971099
HPV53_UC139_LW_V1	*Alphapapillomavirus*	*Alphapapillomavirus 6*	human papillomavirus 53	OP971031
HPV54_UC094_LW_V2	*Alphapapillomavirus*	*Alphapapillomavirus 13*	human papillomavirus 54	OP971049
HPV54_UC137_LW_V2	*Alphapapillomavirus*	*Alphapapillomavirus 13*	human papillomavirus 54	OP971064
HPV56_UC094_LW_V2	*Alphapapillomavirus*	*Alphapapillomavirus 6*	human papillomavirus 56	OP971047
HPV56_UC159_LW_V2	*Alphapapillomavirus*	*Alphapapillomavirus 6*	human papillomavirus 56	OP971068
HPV56_UC091_LW_V1	*Alphapapillomavirus*	*Alphapapillomavirus 6*	human papillomavirus 56	OP971017
HPV56_UC091_LW_V2	*Alphapapillomavirus*	*Alphapapillomavirus 6*	human papillomavirus 56	OP971020
HPV58_UC091_LW_V1	*Alphapapillomavirus*	*Alphapapillomavirus 9*	human papillomavirus 58	OP971016
HPV58_UC164_LW_V1	*Alphapapillomavirus*	*Alphapapillomavirus 9*	human papillomavirus 58	OP971037
HPV58_UC164_LW_V2	*Alphapapillomavirus*	*Alphapapillomavirus 9*	human papillomavirus 58	OP971040
HPV61_UC094_LW_V2	*Alphapapillomavirus*	*Alphapapillomavirus 3*	human papillomavirus 61	OP971045
HPV61_UC094_LW_V3	*Alphapapillomavirus*	*Alphapapillomavirus 3*	human papillomavirus 61	OP971077
HPV61_UC137_LW_V2	*Alphapapillomavirus*	*Alphapapillomavirus 3*	human papillomavirus 61	OP971059
HPV62_UC093_LW_V1	*Alphapapillomavirus*	*Alphapapillomavirus 3*	human papillomavirus 62	OP971022
HPV66_UC115_LW_V3	*Alphapapillomavirus*	*Alphapapillomavirus 6*	human papillomavirus 66	OP971029
HPV67_UC094_LW_V2	*Alphapapillomavirus*	*Alphapapillomavirus 9*	human papillomavirus 67	OP971075
HPV67_UC096_LW_V1	*Alphapapillomavirus*	*Alphapapillomavirus 9*	human papillomavirus 67	OP971025
HPV68_UC094_LW_V2	*Alphapapillomavirus*	*Alphapapillomavirus 7*	human papillomavirus 68	OP971073
HPV68_UC094_LW_V3	*Alphapapillomavirus*	*Alphapapillomavirus 7*	human papillomavirus 68	OP971082
HPV69_UC094_LW_V2	*Alphapapillomavirus*	*Alphapapillomavirus 5*	human papillomavirus 69	OP971051
HPV69_UC094_LW_V3	*Alphapapillomavirus*	*Alphapapillomavirus 5*	human papillomavirus 69	OP971085
HPV69_UC137_LW_V2	*Alphapapillomavirus*	*Alphapapillomavirus 5*	human papillomavirus 69	OP971065
HPV73_UC094_LW_V2	*Alphapapillomavirus*	*Alphapapillomavirus 11*	human papillomavirus 73	OP971050
HPV73_UC119_LW_V2	*Alphapapillomavirus*	*Alphapapillomavirus 11*	human papillomavirus 73	OP971057
HPV74_UC096_LW_V2	*Alphapapillomavirus*	*Alphapapillomavirus 10*	human papillomavirus 74	OP971090
HPV74_UC015_LW_V3	*Alphapapillomavirus*	*Alphapapillomavirus 10*	human papillomavirus 74	OP971000
HPV74_UC096_LW_V3	*Alphapapillomavirus*	*Alphapapillomavirus 10*	human papillomavirus 74	OP971028
HPV81_UC113_LW_V2	*Alphapapillomavirus*	*Alphapapillomavirus 3*	human papillomavirus 81	OP971096
HPV81_UC091_LW_V2	*Alphapapillomavirus*	*Alphapapillomavirus 3*	human papillomavirus 81	OP971019
HPV81_UC091_LW_V3	*Alphapapillomavirus*	*Alphapapillomavirus 3*	human papillomavirus 81	OP971021
HPV81_UC137_LW_V2	*Alphapapillomavirus*	*Alphapapillomavirus 3*	human papillomavirus 81	OP970966
HPV81_UC164_LW_V1	*Alphapapillomavirus*	*Alphapapillomavirus 3*	human papillomavirus 81	OP971035
HPV84_UC137_LW_V2	*Alphapapillomavirus*	*Alphapapillomavirus 3*	human papillomavirus 84	OP971060
HPV84_UC015_LW_V3	*Alphapapillomavirus*	*Alphapapillomavirus 3*	human papillomavirus 84	OP970998
HPV87_UC094_LW_V3	*Alphapapillomavirus*	*Alphapapillomavirus 3*	human papillomavirus 87	OP971052
HPV87_UC055_LW_V1	*Alphapapillomavirus*	*Alphapapillomavirus 3*	human papillomavirus 87	OP971001
HPV87_UC062_LW_V3	*Alphapapillomavirus*	*Alphapapillomavirus 3*	human papillomavirus 87	OP971004
HPV89_UC137_LW_V2	*Alphapapillomavirus*	*Alphapapillomavirus 3*	human papillomavirus 89	OP971058
HPV89_UC096_LW_V1	*Alphapapillomavirus*	*Alphapapillomavirus 3*	human papillomavirus 89	OP971023
HPV90_UC096_LW_V2	*Alphapapillomavirus*	*Alphapapillomavirus 14*	human papillomavirus 90	OP971055
HPV90_UC113_LW_V1	*Alphapapillomavirus*	*Alphapapillomavirus 14*	human papillomavirus 90	OP971094
HPV90_UC084_LW_V1	*Alphapapillomavirus*	*Alphapapillomavirus 14*	human papillomavirus 90	OP971005
HPV90_UC084_LW_V2	*Alphapapillomavirus*	*Alphapapillomavirus 14*	human papillomavirus 90	OP971010
HPV91_UC113_LW_V1	*Alphapapillomavirus*	*Alphapapillomavirus 8*	human papillomavirus 91	OP971095
HPV91_UC113_LW_V2	*Alphapapillomavirus*	*Alphapapillomavirus 8*	human papillomavirus 91	OP971097
HPV108_UC137_LW_V2	*Gammapapillomavirus*	*Gammapapillomavirus 6*	human papillomavirus 108	OP971066
HPV108_UC091_LW_V1	*Gammapapillomavirus*	*Gammapapillomavirus 6*	human papillomavirus 108	OP971018
HPV214_UC113_LW_V2	*Gammapapillomavirus*	unclassified	human papillomavirus 214	OP971056
HPV215_UC145_LW_V1	*Gammapapillomavirus*	unclassified	human papillomavirus 215	OP971101
HPV226_UC084_LW_V2	*Gammapapillomavirus*	unclassified	human papillomavirus 226	OP971015
HPVmSD2_UC094_LW_V3	*Gammapapillomavirus*	unclassified	human papillomavirus mSD2	OP970967

**Table 3 viruses-15-00758-t003:** Mixed effect logistic regression to identify associations between having any HPV, multiple HPVs, any high-risk HPVs, or any low-risk HPVs and cervicovaginal cytokines.

Cytokine	Any HPV		Multiple HPVs		Any High-Risk HPV		Any Low-Risk HPV		Bacterial Vaginosis	
	Beta-Estimate (95% CI)	*p*	Beta-Estimate (95% CI)	*p*	Beta-Estimate (95% CI)	*p*	Beta-Estimate (95% CI)	*p*	Beta-Estimate (95% CI)	*p*
IL-1β	0.20 (−0.36; 0.77)	0.483	−0.41 (−1.22; 0.39)	0.312	0.04 (−0.53; 0.62)	0.895	−0.03 (−0.61; 0.55)	0.921	1.00 (0.30; 1.69)	0.006
IL-6	0.28 (−0.59; 1.16)	0.517	−0.42 (1.45; 0.62)	0.430	−0.16 (−1.02; 0.71)	0.726	0.50 (−0.38; 1.37)	0.268	0.61 (−0.18; 1.40)	0.134
IL-17A	0.42 (−1.01; 1.85)	0.565	−0.48 (−2.12; 1.17)	0.569	−0.06 (−1.49; 1.36)	0.930	1.15 (−0.52; 2.84)	0.178	0.46 (−0.82; 1.75)	0.478
IL-17F	0.48 (−0.47; 1.42)	0.322	−0.33 (−1.33; 0.67)	0.518	0.11 (−0.79; 1.01)	0.812	0.35 (−0.55; 1.25)	0.447	0.82 (0.01; 1.64)	0.048
IL-21	0.03 (−0.32; 1.33)	0.978	−0.05 (−0.79; 0.68)	0.884	0.03 (−0.64; 0.71)	0.927	0.24 (−0.46; 0.95)	0.508	0.11 (−0.47; 0.71)	0.707
IL-22	0.17 (−1.28; 1.62)	0.817	−1.48 (−3.76; 0.79)	0.200	−0.52 (−2.11, 1.07)	0.523	0.34 (−1.08; 1.75)	0.641	0.28 (−1.02; 1.57)	0.676
IL-23	0.28 (−0.52; 1.08)	0.491	−0.34 (1.20; 0.52)	0.435	−0.16 (−0.95; 0.62)	0.683	0.65 (−0.31; 1.62)	0.185	0.69 (−0.04; 1.42)	0.063
IL-25	0.08 (−0.65; 0.81)	0.825	−0.59 (−1.38; 0.21)	0.149	−0.33 (−1.04; 0.37)	0.353	0.12 (−0.60; 0.84)	0.741	0.68 (0.03; 1.33)	0.041
IL-31	−0.53 (−2.15; 1.08)	0.517	−1.75 (−3.81; 0.30)	0.094	−1.33 (−3.06; 0.40)	0.132	0.52 (−1.03; 2.07)	0.510	0.81 (−0.56; 2.18)	0.249
IL-33	0.27 (−0.92; 1.45)	0.661	−0.43 (−1.72; 0.84)	0.502	−0.42 (−1.60; 0.76)	0.486	0.61 (−0.68; 1.90)	0.355	1.34 (0.12; 2.57)	0.032
IFN-γ	−0.50 (−1.92; 0.91)	0.484	−1.60 (−3.27; 0.07)	0.061	−1.22 (−2.76; 0.31)	0.118	0.06 (−1.22; 1.34)	0.928	1.10 (−0.22; 2.43)	0.103
sCD40L	0.08 (−0.61; 0.76)	0.828	−0.47 (−1.26; 0.32)	0.240	−0.31 (−1.01, 0.39)	0.390	−0.07 (−0.72; 0.59)	0.838	0.34 (−0.23; 0.96)	0.223
TNF-α	0.26 (−0.58; 1.10)	0.552	−0.58 (−1.54; 0.38)	0.236	−0.11 (−0.92; 0.70)	0.788	−0.24 (−1.06; 0.58)	0.560	1.60 (0.52; 2.68)	0.004

## Data Availability

Papillomaviruses sequences have been deposited in GenBank under accessions OP970964-OP971102, and raw reads have been deposited in SRA under BioProjects PRJNA767784 and PRJNA881266.

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
