# Peer review of "Cervicovaginal Human Papillomavirus Genomes, Microbiota Composition and Cytokine Concentrations in South African Adolescents"

_viruses, 2023, doi:10.3390/v15030758_

Round 1

Reviewer 1 Report

Dear Authors,

Thank you for allowing me to review your manuscript.

The study is very interesting and I really congratulate you on this research.

The introduction is well documented and the purpose of the study is clearly expressed.

At result section some correlations are always a good point.

Discussion and conclusions are well written.

In addition, I would reserve a paragraph dedicated to possible clinical developments in humans and another to the limits of the study that are not adequately described.

It would be important to carry out further comparisons and insights on other populations and to evaluate the variability of the microbiota resulting from different diets in different populations.

Finally, for a multidisciplinary approach we propose to mention:

-DOI: 10.1186/s13027-022-00465-9

-DOI: 10.3390/jpm12091387 

- DOI: 10.3390/biology11081114

Author Response

Dear Authors,

Thank you for allowing me to review your manuscript.

The study is very interesting and I really congratulate you on this research.

The introduction is well documented and the purpose of the study is clearly expressed.

At result section some correlations are always a good point.

Discussion and conclusions are well written.

In addition, I would reserve a paragraph dedicated to possible clinical developments in humans and another to the limits of the study that are not adequately described.

It would be important to carry out further comparisons and insights on other populations and to evaluate the variability of the microbiota resulting from different diets in different populations.

Finally, for a multidisciplinary approach we propose to mention:

-DOI: 10.1186/s13027-022-00465-9

-DOI: 10.3390/jpm12091387 

- DOI: 10.3390/biology11081114

Thank you, we appreciate your comments. We have added a paragraph discussing possible clinical developments in humans and have extended the limitations paragraph, as suggested. We also included the references listed.

Reviewer 2 Report

In this paper, the authors explore the interactions between vaginal virome, bacteriome, and the profile of cytokines in females from South Africa. They found coinfection with several HPV types in multiple samples, correlation with dysbiosis and finally increase in some pro-inflammatory cytokines but without statistically significant.

I have some concerns that should be considered.

1.       The authors do not mention in the introduction, why is the analysis particularly important in this age group, when many cervical cancer cases occur in advanced ages.

2.       The author observed that “The proportion of women with high-risk HPV types was 63.2% (12/19) for participants with BV and 45.5% (10/22) for women without BV”, but it is not clear what is being evaluated with the p value= 0.350. If a significant difference between these groups was evaluated, it was not mentioned in the paragraph.

3.       Several correlation analyses were done but the absence of values raises the question: When a positive or negative correlation is significant?

4.       In addition to encouraging vaccination, could some of the identified correlations have an impact on prioritizing treatments, emphasizing follow-up, or otherwise impacting the care provided to patients? It should be addressed in the discussion.

5.       The “PID” meaning is not described and along with participant ID the terms are employed indistinctly which difficulted the reading. The spelling needs to be revisited.

Author Response

In this paper, the authors explore the interactions between vaginal virome, bacteriome, and the profile of cytokines in females from South Africa. They found coinfection with several HPV types in multiple samples, correlation with dysbiosis and finally increase in some pro-inflammatory cytokines but without statistically significant.

I have some concerns that should be considered.

  1. The authors do not mention in the introduction, why is the analysis particularly important in this age group, when many cervical cancer cases occur in advanced ages.

Recently, HPV infections have been linked to increased HIV risk. Given that young women are at the highest risk of HIV acquisition in Sub-Saharan Africa, this analysis is of importance for this age group. We have clarified this in the introduction.

  1. The author observed that “The proportion of women with high-risk HPV types was 63.2% (12/19) for participants with BV and 45.5% (10/22) for women without BV”, but it is not clear what is being evaluated with the p value= 0.350. If a significant difference between these groups was evaluated, it was not mentioned in the paragraph.

We have compared the proportion of women with high-risk HPV types in women with vs those without BV. This was not significant, which we have clarified in the text.

  1. Several correlation analyses were done but the absence of values raises the question: When a positive or negative correlation is significant?

We have clarified that the correlations between the presences of specific HPV types was based on co-occurrence and listed the significant co-occurrences in the text. We used the R package cooccur for this analysis. The algorithm calculates the observed and expected frequencies of co-occurrence between each pair of HPV types. The expected frequency is based on the distribution of each HPV type being random and independent of the other types.The analysis returns the probabilities and associated p-values that a more extreme (either low or high) value of co-occurrence could have been obtained by chance.

For the microbiota analysis, we used sPLSDA to identify bacterial taxa that accounted for the highest degree of variance between women with and without any HPV and included only those with significant values in subsequent analyses. The network analysis thus only includes taxa that showed significant associations. We have added this to the text.

  1. In addition to encouraging vaccination, could some of the identified correlations have an impact on prioritizing treatments, emphasizing follow-up, or otherwise impacting the care provided to patients? It should be addressed in the discussion.

 Thank you, we have added a paragraph on potential implications.

  1. The “PID” meaning is not described and along with participant ID the terms are employed indistinctly which difficulted the reading. The spelling needs to be revisited.

Thank you, we have clarified this and now use one term consistently in the text and figure legends.